# Autoimmune Diseases: Enzymatic cross Recognition and Hydrolysis of H2B Histone, Myelin Basic Protein, and DNA by IgGs against These Antigens

**DOI:** 10.3390/ijms23158102

**Published:** 2022-07-22

**Authors:** Georgy A. Nevinsky, Valentina N. Buneva, Pavel S. Dmitrienok

**Affiliations:** 1Institute of Chemical Biology and Fundamental Medicine of the Siberian, Division of Russian Academy of Sciences, Lavrentiev Ave. 8, 630090 Novosibirsk, Russia; buneva@niboch.nsc.ru; 2Pacific Institute of Bioorganic Chemistry, Far East Division, Russian Academy of Sciences, 690022 Vladivostok, Russia; paveldmt@piboc.dvo.ru

**Keywords:** human blood sera antibodies-abzymes, HIV-infected and multiple sclerosis patients, catalytic abzymes, hydrolysis of H2B histone, IgGs against H2B, H1, H2A, H3, H4 histones, myelin basic protein, DNA, enzymatic cross recognition and hydrolysis

## Abstract

As shown in many studies, one of the earliest statistically significant indicators of the development of many autoimmune diseases (ADs) is the appearance in the blood of antibodies with catalytic activities (abzymes) hydrolyzing different autoantigens. Antibodies-abzymes having different enzymatic activities are a specific and essential feature of some ADs. Most abzymes are harmful to humans. Free histones in the blood are damage-associated proteins, and their administration to animals drives systemic inflammatory and toxic effects. Myelin basic protein (MBP) is the most critical component of the axon myelin-proteolipid sheath. Hydrolysis of MBP by abzymes leads to the disruption of nerve impulses. Here, we analyzed the possible pathways for the formation of unusual antibodies and abzymes that exhibit polyspecificity in recognition during complex formation with partially related antigens and possess the ability to catalyze several different reactions for the first time. Using IgGs of HIV-infected and multiple sclerosis patients against five individual histones (H1–H4), MBP, and DNA, it was first shown that abzymes against each of these antigens effectively recognize and hydrolyze all three antigens: histones, MBP, and DNA. The data obtained indicate that the formation of such polyspecific abzymes, whose single active center can recognize different substrates and catalyze several reactions, can occur in two main ways. They can be antibodies against DNA–protein complex hybrid antigenic determinants containing proteins and nucleic sequences. Their formation may also be associated with the previously described phenomenon of IgG extensive LH half-molecule (containing one L-light and one H-heavy chains) exchange leading to H_2_L_2_ molecules containing HL halves with variable fragments recognizing different antigens.

## 1. Introduction

Antibodies (Abs) to chemically stable analogs of various reaction transition states and natural auto-antibodies with enzymatic activities are called abzymes (ABZs); they are well-described in the literature [1,2,3,4,5,6]. The development of several pathologies associated with contravention of the immune system, including some autoimmune diseases (ADs) and some viral diseases, results in the synthesis of abzymes by B-cells against polysaccharides, lipids, peptides, proteins, DNAs, RNAs, and their complexes [2,3,4,5,6]. In the blood sera of such patients, there may be many different ABZs against various specific antigens that mimic chemical reactions transition states. Catalytically active secondary anti-idiotypic auto-antibodies to active sites of some classical enzymes were also found, the formation of which may be explained using Jerne’s model of the anti-idiotypic network [7]. The appearance of immunoglobulins (Igs) with catalytic activities in the blood sera of sick mammals is statistically dependable and the earliest indicator of the onset of many autoimmune disorders [2,3,4,5,6]. To date, abzymes (IgGs, IgA, and IgMs) degrading DNAs, RNAs [8,9,10,11,12], poly and oligosaccharides [13,14,15], various peptides, and proteins [16,17,18,19,20,21,22,23] have been found in the blood sera of patients with different ADs and several viral pathologies [2,3,4,5,6].

Antibodies to various auto-antigens are produced in the blood of not only sick but also conventionally healthy humans [2,3,4,5,6]. Some healthy people sometimes have antibodies with shallow activity, splitting vasoactive intestinal peptides [16], thyroglobulin [18], or polysaccharides [13,14,15]. At the same time, the blood of healthy people and patients suffering from some pathologies with insignificant autoimmune reactions usually lack ABZs [2,3,4,5,6].

The question concerning abzymes and their biological role is critical for understanding the mechanisms of many Ads’ development. To obtain monoclonal light chains with DNase and myelin basic protein (MBP)-hydrolyzing activities, the cDNA kappa library of light chains of peripheral blood antibodies of three SLE patients (10^6^ variants of different light chains) and the phage display method were used [24,25,26,27,28,29,30,31,32]. To obtain individual colonies corresponding to DNase IgGs, phage particles eluted from DNA cellulose with 0.5 M NaCl and acidic buffer (pH 2.6) were used [24,25]. Forty-five out of four hundred and fifty-one individual colonies were randomly selected, and fifteen out of forty-five were selected for analysis of phage particle preparations, and corresponding antibodies (~33%) efficiently hydrolyzed DNA, demonstrating very different enzymatic properties [24]. A total of 33 out of 687 individual phage particles eluted from DNA cellulose with acidic buffer were selected. Nineteen out of thirty-three clones (58%) and their corresponding antibodies exhibited DNase activity [25]. In contrast to canonical DNases, all 34 preparations of monoclonal light chains (MLChs) differed in their level of activation by various metal ions (Na^1+^, K^1+^, Mg^2+^, Mn^2+^, Co^2+^, Ni^2+^, Ca^2+^). All MLCh preparations were characterized by different optimal pHs and apparent reaction constants (*k*_cat_) [24,25].

A pool of phage particles containing MLChs with different affinities for MBP was divided into ten fractions using affinity chromatography on MBP-Sepharose [26,27,28,29]. In total, 72 colonies were randomly selected from 440 individual colonies. Of the 72 colonies, 22 (~30%) had MBP-hydrolyzing activity [26]. This showed that 12 of 22 MLChs are metalloproteases; their activities were suppressed only by EDTA. Four MLChs were serine-like proteases; only PMSF repressed their activity. The effects of PMSF and EDTA in the case of three MLChs were comparable: ~40% and 60%, respectively. The activity of three MLChs was strongly suppressed only by iodoacetamide, a specific inhibitor of thiol-proteases. Interestingly, iodoacetamide significantly inhibited the activity of two MLChs, which were also significantly inhibited by EDTA [26]. It was logical to assume that several MLChs are chimeric Igs, the active centers of which contain amino acid (AA) residues of serine, thiol, and metalloproteases [26]. These data indicated an exceptional variety of dependences of the activities of different MLChs on ions of various metals, pH optima, and *k*_cat_ in MBP hydrolysis [26].

Three additional MLChs have been analyzed in more detail [27,28,29]. DNA sequences of these MLChs were studied; they were identical (88–100%) to the germlines of IgLV8 light chain genes of several described antibodies [27,28,29]. NGTA1-Me-pro was a typical metalloprotease [27]. It had two different very well-pronounced optimum pHs (6.0 and 8.5), two optimal concentrations of CaCl_2_ (1.0 and 6.0 mM), two *K*_m_ values for MBP, and two *k*_cat_ at different pHs and concentrations of CaCl_2_. These data unambiguously indicated that NGTA1-Me-pro has two metal-dependent active sites in one active center [27]. The sequence of NGTA1-Me-pro was shown to have a double set of protein sequences characteristic of the recognition sites of proteins, metal ions, and catalytic amino acid residues necessary for the hydrolysis of proteins in the case of several canonical metal proteases [30]. Specific inhibitors of NGTA2-Me-pro-Tr were PMSF (42%) and EDTA (58%): it exhibits the properties of a chimeric protease with serine and metal-dependent activities [31]. The pH optimum for metalloprotease activity was 6.5, while its serine-like activity had an optimum of pH 7.5. Thus, NGTA2-Me-pro-Tr was the first example of an MLCh having two combined centers with serine-like and metalloprotease activities [28]. DNA analysis of the NGTA2-Me-pro-Tr sequence showed that it contains specific protein sequences, which, in the case of serine-like and metalloproteases, are responsible for recognizing protein substrates and chelation of metal ions as specific cofactors of classical metalloproteases [31]. In addition, the active center of NGTA2-Me-pro-Tr contains amino acid residues responsible for the hydrolysis of proteins in canonical serine and metalloproteases.

It should emphasize that all recombinant MLChs were obtained by affinity chromatography of phage particles on MBP-Sepharose. Considering this, a very unexpected result was obtained for NGTA3-pro-DNase [29]. Western blotting showed that NGTA3-pro-DNase gives a positive response not only to MBP but also to DNA. After SDS-PAGE, the positions of MBP-hydrolyzing serine, metalloprotease, and DNase activities coincided and corresponded only to gel fragments containing MLChs [29]. These MLChs have an optimum pH of their metalloprotease activity at 8.6 and in serine-like activity at pH 7.0. Due to the serine-like activity, MLChs catalyzed MBP hydrolysis two times faster than metalloprotease activity [29]. In addition to two protease activities, NGTA3-pro-DNase possesses DNase activity. It has a pH optimum of 6.5 and affinity for DNA about 3.5 orders of magnitude higher than DNase I [29]. The sequence analysis allowed identifying fragments of protein sequences of NGTA3-pro-DNase responsible for the binding of MBP, DNA, and two clusters responsible for chelation of metal ions, AA residues of three types of active centers involved directly in the catalysis of two proteases and DNase activities [32]. Computer simulation has shown that NGTA3-pro-DNase has good structural similarity with a crystal structure of a catalytic antibody having a serine protease active site; the difference is 1.79 Ȧ, with the crystal structure of HIV-1 neutralizing Abs in complex oligopeptide and oligonucleotide [32].

For classical enzymes, the situation is quite simple: one gene, one enzyme. There are no examples of canonical enzymes when the same active site can catalyze two or more different reactions. The above-mentioned data show that monoclonal abzymes with the same catalytic function can be very different. Due to the V (D) J recombination, unique DNA regions encoding variable domains of antibodies may be formed. The variable regions of heavy (H) or light (L) chains are encoded by a locus divided into several V, D, and J fragments [33,34,35,36]. From a theoretical point of view, the human immune system can create one antigen against about 10^5^–10^6^ B-lymphocytes, which can produce Abs to the same antigen with different properties. This is in good agreement with our experimental data [24,25,26,27,28,29,30,31,32].

In described studies, a cDNA library of only kappa light chains of Abs from patients with SLE was used [24,25,26,27,28,29,30,31,32]. Only about 10% of individual colonies were analyzed for DNase activity, and from 33% [24] to 58% [25] of analyzed homogeneous MLChs exhibited DNase activity. Only 16% of individual colonies were used to analyze MBP-hydrolyzing activity, and ~35% of them produced IgGs hydrolyzing MBP [26,27,28,29,30,31,32]. At the same time, both kappa and lambda Abs have DNase and MBP-hydrolyzing activities [2,3,4,5,6,36]. If we take into account the average value of the percentage of active abzymes in one peak, equal to about 42%, and the 10 peaks obtained under affinity chromatographies, all of which contain abzymes, as well as the percentage of individual colonies analyzed, the possible number of kappa abzymes with DNase and MBP-hydrolyzing activities can be ≥1000, and taking into account lambda-Abs, this can be even more [24,25,26,27,28,29,30,31,32,37]. These data show one of the pathways for forming an exceptional variety of antibodies with and without catalytic activity. However, there is another way of creating polyfunctional intact autoantibodies and abzymes.

In the classic paradigm, the clonal B-cell populations produce IgGs recognizing a single antigen, and they are L_2_H_2_ molecules, two LH parts of which have two identical antigen-binding sites. However, in human blood, milk, and placenta, IgGs to various antigens undergo extensive LH half-molecule exchange [38,39,40]. In the IgG pool, independently of sources, in the range 33.0–62.4% and 13.0–29.8% of Abs contained L chains exclusively of κ-κ or λ-λ type, respectively, while 8.8–54.0% of the chimeric IgGs contained both kappa- and lambda-light chains [38,39,40]. Chimeric κ-λ-IgGs consisted of different amounts of HL halves of IgG1, IgG2, IgG3, and IgG4 [38,39,40]. A similar situation was observed for sIgAs from human milk [41]. As a result of the exchange, all IgG fractions were eluted from every one of several specific affinity sorbents under severe conditions, destroying strong immunocomplexes showing high catalytic activities in the hydrolysis of several substrates (ATP, DNA, oligosaccharides) and other chemical reactions [38,39,40,41]. Thus, the same complete H_2_L_2_ molecules of Abs can hydrolyze several different substrates, which depend on against which antigens HL halves entered the molecule due to the exchange.

As indicated above, the active center of monoclonal NGTA3-pro-DNase chains combines three active centers: serine-like, metalloprotease, and DNase [29,32]. It seemed interesting against which antigen Abs can be produced to hydrolyze both proteins and DNA. It showed that the main antigen against which anti-DNA Abs are produced during ADs are DNA complexes with histones, which appear in the blood due to cell apoptosis [42]. About 30–40% of these antibodies from patients with ADs have DNase activity [2,3,4,5,6,24,25]. In addition, the blood of healthy donors contains Abs against histones, and in patients with ADs, there are also abzymes hydrolyzing all five histones [43,44]. In general, the situation with the recognition of different molecules and the catalysis of their further transformation in the case of antibodies-abzymes can be very different from classical enzymes. Analysis of the processes of recognition of substrates and catalysis of reactions is a new direction in biochemistry and enzymology.

It could be assumed that in the case of DNA–histone complexes, antibodies can be produced not only against DNA or histones but also against antigenic determinants formed at the junction of protein sequences and fragments of DNA bound with histones. Taking this into account, this hypothesis was tested in this work. For this, specific IgG antibodies against histones and DNA have been obtained, and an analysis of whether Abs against histones can hydrolyze DNAs and vice versa was performed. It has been shown that IgG antibodies against DNA and several histones from the sera of MS and HIV-infected patients can recognize and catalyze cross-hydrolysis of H2B histone, MBP, and DNA.

## 2. Results

### 2.1. Purification of Antibodies

It previously shown that IgGs from the blood of MS and HIV-infected patients hydrolyze all five histones and MBP [43,44,45,46,47,48,49,50,51]. In addition, it was found that HIV-infected patients’ antibodies against five histones have the ability to recognize and hydrolyze not only five histones but also MBP and vice versa; IgGs against MBP hydrolyze histones [48,49,50]. Moreover, IgGs from MS patients against five histones have been shown to hydrolyze the H1 histone effectively [51]. As mentioned above, monoclonal light chains of SLE patients can be contained in one site from one to three active centers corresponding to different classical enzymes. The question of the specific pathways and mechanisms by which AA residues corresponding to several different canonical enzymes can occur in the same active site is of particular interest. The main goal of this work was to analyze the possibility of the formation of antibodies against combined protein–DNA antigenic determinants at the junction of histones and DNA in their complexes, which appear in the blood as a result of cell apoptosis. In the case of the formation of such antibodies, their active centers could contain structural elements responsible for the catalysis of proteins and DNA hydrolysis. For this purpose, by analogy with [48,49,50,51], homogeneous preparations of antibodies of MS patients against five histones, MBP, and DNA containing no classical proteases and DNases were obtained for the first time. In addition, in this work, preparations of antibodies of HIV-infected against DNA were also obtained for the first time. Polyclonal IgGs against five histones and MBP were removed from the total preparations of polyclonal antibodies of HIV-infected and MS patients by their affinity chromatography on MBP- and histone-5-Sepharose. Antibodies eluted from these two sorbents during loading were additionally passed through these sorbents and then applied to DNA–cellulose. Antibodies with a high affinity for DNA–cellulose were eluted from this sorbent with 3.0 M NaCl and acidic buffer and used as anti-DNA IgGs.

### 2.2. DNase Activity of IgGs against DNA and Proteins

First, an analysis was performd of the possibility of DNA hydrolysis with antibodies against DNA, five histones, and MBP corresponding to ms-IgG_mix_ and hiv-IgG_mix_ (mixtures of seven IgG preparations in each case, see above). As an example, Figure 1 shows the data of the analysis of supercoiled (sc)DNA hydrolysis by several IgGs.

Hydrolysis of the substrate can occur only after its recognition by enzymes/abzymes and the formation of a specific complex. It can be seen that, after 3 h of incubation with IgGs (10 µg/mL) against the DNA of MS and HIV-infected patients, scDNA is completely hydrolyzed to the linear form of DNA (two or more breaks per molecule) and oligonucleotides of different lengths. It is very important that IgGs against five histones corresponding to MS and HIV-infected patients effectively hydrolyzed scDNA during this time by about 25–35% with the formation of a relaxed (one break per molecule) and linear form of DNA. Moreover, IgGs of MS and HIV-infected patients against MBP also hydrolyzed DNA by 20–35%. An increase in the concentration of IgGs against five histones and MBP and the incubation time led to the formation of short hydrolysis products, as in the case of Abs against DNA. These data indicated that, as in the case of monoclonal antibodies against MBP obtained by Phage display [29], the active centers of some monoclonal antibodies in the fractions of polyclonal IgGs of HIV-infected and MS patients against histones and MBP could combine at least two activities—protease and DNase.

### 2.3. Protease Activity of IgGs against Histones and MBP

It was shown that antibodies of HIV-infected patients efficiently hydrolyze all human histones [43]. In addition, IgGs of HIV-infected patients against MBP and five histones possess polyspecific complex recognition and enzymatic cross-reactivity in the hydrolysis of five histones and MBP [48,49,50]. Hiv-IgG against five histones and MBP used in this study also hydrolyzed five histones (Figure 2A,B). In this work, we have shown for the first time that IgGs of MS patients against five histones split five histones (Figure 2A,B).

Electrophoretically homogeneous preparations of MBP, unfortunately, are not available. Due to multiple splicing of cDNA and fractional hydrolysis of MBP in some human brains, MBP preparations could contain several protein forms (18.5, 17.5, ≤14.0 kDa) and products of their hydrolysis [52]. Lane C (Figure 2B) shows the MBP starting preparation heterogeneity containing mainly the 18.5 kDa protein form. After 15 h of MBP incubation with IgG_mix_ of HIV-infected and MS patients against five histones and MBP, the 18.5 kDa MBP form decreases greatly compared to the control (lane C) (Figure 2B). In all cases, the formation of significantly smaller proteins and short peptides is observed. This may indicate that not only HIV-infected IgGs against MBP and histones [48,49,50,51] but also those of MS patients possess enzymatic cross-reactivity in the hydrolysis of histones and MBP.

Figure 3 shows data on the absence of hydrolysis by antibodies against five histones and MBP of eight control proteins; these antibodies cross-hydrolyze only MBP and histones (Figure 2).

### 2.4. Catalytic Cross-Reactivity of IgGs against Histones, MBP, and DNA

As described above, we obtained antibodies of MS patients against five individual histones (H1–H4). It was interesting to evaluate the level of the relative activity of these IgGs in the hydrolysis of histones. Figure 4 demonstrates data on the hydrolysis of five histones with antibodies against five individual histones: anti-H1, anti-H2A, anti-H2B, anti-H3, anti-H4, and anti-DNA IgGs.

It can be seen that all IgG preparations hydrolyze five histones to varying degrees. The maximum activity in the hydrolysis of all five histones is demonstrated by antibodies against histone H1 (line H1). A particular question was whether IgG antibodies against DNA, which have a high affinity for DNA cellulose, are capable of hydrolyzing five histones. It turned out that IgGs of HIV-infected (lane DNA1) and MS (lane DNA2) patients also effectively hydrolyze all five histones (Figure 4). Together with the results of DNA hydrolysis with IgGs against five histones (Figure 1), these data indicated that the catalytic centers of antibodies against histones might contain specific structural elements of canonical DNases, while the active centers of anti-DNA IgGs may contain amino acid residues of canonical proteases. A feature of IgGs against histones and MBP is that, unlike canonical proteases, they hydrolyze only histones and MBP, but not other control proteins [48,49,50].

### 2.5. MALDI Mass Analysis of Catalytic Cross-Reactivity

The fractions of IgGs with high affinity to five individual histones (H1–H4), MBP and DNA were used to find the cleavage sites of H2B by MALDI TOFF mass spectrometry. After the addition of the IgGs at a time of zero (Figure 5A), the H2B histone was nearly homogeneous, demonstrating only signals of its one- (m/z = 13,780.6 Da) and two-charged ions (m/z = 6890.3 Da).

One of the exciting queries was whether IgGs of MS patients against H2B, H1, H2A, H3 and H4 histones could hydrolyze the H2B histone. H2B cleavage assays were carried out with IgGs against five histones and DNA after 3–24 h of incubation. Not all peaks corresponding to various H2B histone splitting sites by all IgGs appeared after 3–6 h of hydrolysis, but all products were detected after 24 h of incubation (54). Based on the analysis of peaks in 9–10 spectra corresponding to a different incubation time, 11 sites of H2B hydrolysis were identified in the case of IgGs against H2B and H2A, and 12 and 7 for anti-H3 and anti-H4 Abs, respectively. All sites of H2B hydrolysis by all these IgGs are shown in Figure 6. A very unexpected result was obtained after incubation of H2B with MS IgGs against histone H1. Anti-H1 IgGs very weakly hydrolyzed H2B even after 24 h of incubation.

Assuming, in the case of complexes of DNA with histones, the possibility of the formation of antigenic determinants containing fragments of proteins and DNAs, one could expect that in the total pool of antibodies, there may be IgGs whose active centers contain amino acid residues of both proteases and DNases. As mentioned above, before the affinity chromatography of IgGs on DNA cellulose, we carefully removed antibodies against five histones and MBP from departure preparations. However, anti-DNA antibodies hydrolyzed the H2B histone very efficiently to form a large number of products (Figure 5F). If in the case of IgGs against four histones 7–12, for anti-DNA antibodies, 31 hydrolysis sites were revealed (Figure 6). While IgGs against histones cleaved H2B at only 1–4 major sites, anti-DNA Abs demonstrated 11 major sites (Figure 6). All sites of H2B hydrolysis by Abs against four different histones are located mainly in specific clusters. At the same time, there is no significant coincidence of the hydrolysis sites of H2B with antibodies against different individual histones. Of the 31 sites of hydrolysis of H2B with anti-DNA-ms-IgGs and all anti-histone antibodies, the following numbers of sites coincide: three (anti-H4), eight (anti-H2B and anti-H2A), and nine (anti-H3) (Figure 6). In addition, the coincident sites of hydrolysis in the case of IgGs against different histones and anti-DNA for various antibodies are different. Moreover, some major sites of H2B hydrolysis by anti-DNA IgGs in the case of Abs against histones are overage or minor. Overall, 15 of 31 sites of H2B hydrolysis with anti-DNA-ms-IgGs were found only for anti-DNA IgGs (Figure 6).

The polyspecificity in binding recognition and catalytic cross-reactivity of IgGs against various histones from the blood of HIV-infected patients was investigated using MALDI mass spectroscopy earlier in several studies [48,49,50]. In this work, IgGs were isolated from the blood of HIV-infected patients against H2B histone, MBP and DNA. It was interesting to understand whether HIV-infected patients’ antibodies against H2B histone and MBP are capable of hydrolyzing H2B and DNA and vice versa. Figure 7 shows the spectra of the products of H2B hydrolysis by antibodies of HIV-infected patients against H2B, MBP, and DNA.

Based on a set of data from 7–10 spectra after 3–20 h of H2B hydrolysis with these antibodies, the sites of hydrolysis of this histone were found (Figure 8).

To simplify the analysis of common and different hydrolysis sites for all antibodies used, they are summarized in Table 1.

Hydrolysis of H2B by HIV-infected patients’ antibodies against this histone occurs at 11 sites (Figure 8A, Table 1). For hydrolysis of H2B with antibodies against MBP of HIV-infected patients, 10 hydrolysis sites were found (Figure 8B, Table 1). Interestingly, there is no overlap among these sites. For comparison, Table 1 shows the data on the sites of H2B hydrolysis by IgGs against MBP of MS patients. In the case of Abs against MBP of MS patients, 11 hydrolysis sites were found (Figure 8C). The 10 and 11 sites of H2B hydrolysis by IgGs against MBP from the blood of HIV-infected and MS patients (Figure 8B,C) are located mainly in the same clusters of this histone sequence. However, there are also no same sites of H2B cleavage (Figure 8B,C).

As shown above, IgGs against DNA from MS patients effectively hydrolyze H2B at 31 sites (Figure 6E, Table 1). IgGs against DNA of HIV-infected patients cleave H2B at 27 sites (Figure 8D, Table 1). A total of 13 hydrolysis sites are the same for anti-DNA antibodies of HIV-infected and MS patients (Table 1), and they are localized in the same AA clusters of the H2B sequence. (Figure 6E and Figure 8D). Thus, IgGs against five different histones (Figure 6 and Figure 8) possess cross polyspecific recognition–complexation and enzymatic cross-reactivity. Moreover, antibodies with a high affinity for histones and MBP can efficiently hydrolyze DNA and vice versa.

## 3. Discussion

The multispecificity of the complexation of antibodies with foreign molecules (misidentification) has long been described in the literature [53,54,55,56]. Antibodies against specific antigens can effectively recognize molecules structurally similar to the specific antigen. This widespread phenomenon was called polyreactivity or polyspecificity of antibody complex formation [53,54,55,56]. Several studies showed that Abs against some antigens are capable of changing the structure of their binding sites when interacting with molecules partially related to a specific antigen [53,54,55,56]. However, the question remains of whether the only reason for polyreactive complexation is the main reason of enzymatic cross-reactivity. Unfortunately, analysis of only complexation cannot lead to an answer to this question. This is because the efficiency of the formation of antibodies and canonical enzyme complexes with various specific and nonspecific ligands differs by only one to two orders of magnitude [57,58,59,60,61]. In many articles, it has been shown that the efficiency of specific substrate selection by canonical enzymes and antibodies at the stage of primary complex formation is only 1–2 and very rarely provides three orders of magnitude [59,60,61]. Then, there is a conformational change in enzymes, antibodies, and associated substrates. These specific changes in their conformations lead to the adjustment of interacting molecules to their catalytically competent state, which provide “orbital control-fitting” of the reacting groups of substrates and enzymes with an accuracy of 10–15° that is possible only for specific substrates [57,58,59,60]. In general, the specificity of enzyme and abzyme action reaches a high level due to 1–2 orders of magnitude of the complexation stage, and the catalysis provides an increase in specificity due to the rise in the rate of the reaction by 5–8 orders of magnitude in comparison with the nonspecific substrates [57,58,59,60,61,62].

Canonical enzymes catalyze only one chemical reaction. Consequently, in contrast to nonspecific complex formation typical for classical specific enzymes, enzymatic cross-reactivity of canonical enzymes is an exclusively rare case [57,58,59,60,61,62]. With this in mind, it was necessary to understand by which mechanisms the catalytic cross-reactivity of abzymes against different antigens could exist.

Canonical enzymes and abzymes against specific antigens during affinity chromatography can form complexes with the immobilized alien molecules [2,3,4,5,6,53,54,55,56]. However, the affinities of enzymes and abzymes for unspecific substrates are usually 1–2 orders of magnitude lower than for specific ones. Therefore, unspecific proteins can generally be eluted from affinity sorbents using 0.1–0.15 M NaCl [2,3,4,5,6,36,37]. Therefore, for isolating IgGs against histones and MBP, we eluted nonspecifically bound Abs using 0.2 M NaCl. For additional nonspecifically bound Abs against five histones and MBP, they were passed through extra alternative affinity sorbents. Finally, IgG fractions against histones and MBP were obtained. Fractions of total antibodies were used to obtain anti-DNA IgGs after carefully removing Abs against histones and MBP by several repeated passes of these fractions through affinity sorbents with immobilized histones and MBP. In the case of all antibodies against histones, MBP, and DNA, only fractions with a high affinity for these immobilized antigens were used, eluted from columns with 3.0 M NaCl and an acidic buffer, destroying strong immunocomplexes.

It was shown that even before using several additional affinity sorbents, IgG preparations of HIV-infected and MS patients do not contain any classical enzymes [45,46,47,48,49,50]. The same inference could be drawn from H2B cleavage sites against histones, MBP, and DNA. Trypsin splits proteins after the lysine (K) and arginine (R) residues, whereas chymotrypsin is after aromatic amino acids (F, Y, and W). All IgGs’ cleavage sites of H2B occur mainly in clusters after neutral non-charged and nonaromatic amino acids: S, G, E, L, A, P, Q, and T (Figure 6 and Figure 8; Table 1). Thus, most cleavage sites by all IgG preparations do not correspond to chymotrypsin or trypsin and are located mainly in specific AA clusters and not along the entire length of the H2B histone molecules.

Using monoclonal Abs of SLE patients, it was shown that their active centers could combine in one-center AAs of serine and metal-dependent proteases and, in addition, DNases [26,27,28,29,30,31,32]. Considering this, it should be assumed that the catalytic centers of some monoclonal IgGs in the composition of polyclonal Abs of HIV-infected and MS patients, which have a high affinity for individual histones and DNA, could combine the active centers of typical proteases and DNases. This may indicate that antibodies with combined active centers can be produced in patients’ blood against hybrid antigenic determinants of histone–DNA complexes consisting of fragments of histones and DNA sequences. In addition, Abs against different histones, as previously shown, hydrolyze only various histones and MBP, but not other control proteins [43,44,45,46,47,48,49,50,51]. This is because all five histones’ protein sequences have a high level of homology with each other and with MBP [45,46,47,48,49,50,51]. Therefore, antibodies of HIV-infected individuals against MBP effectively hydrolyze all five histones [45,46,47,48,49,50,51]. IgGs against MBP of MS patients efficiently hydrolyze H2B. However, somewhat unexpectedly, among 11 and 10 sites of H2B splitting with IgGs against MBP of HIV-infected and MS patients, there are none of the same sites (Figure 6 and Figure 8; Table 1). This may be because MBP has four antigenic determinants [42], which are, to varying degrees, homologous to the H2B histone protein sequence [49]. It is possible that this histone in the blood of HIV-infected and MS patients is represented by different complexes with DNA and other blood components, including various proteins and DNA or RNA. In this case, the production of abzymes in the blood of HIV-infected and MS patients can occur in different accessible MBP sequences and, as a result, for the hydrolysis of H2B at different sites.

It is not yet fully understood how IgGs with high affinity for MBP are formed, which effectively hydrolyze DNA. On the one hand, it can be any specific Abs against complexes of histones with DNA that have a high affinity for MBP. However, as was shown earlier, MBP, like histones, can form complexes with DNA with a reasonably high affinity [63]. Taking this into account, it cannot be ruled out that IgGs that effectively hydrolyze MBP and DNA are Abs against hybrid protein–nucleic acid antigenic determinants of MBP–DNA complexes. IgGs against DNA from MS patients effectively hydrolyze H2B at 31 sites, while HIV-infected patients did so at 27 sites (Figure 6 and Figure 8; Table 1). Only 13 hydrolysis sites are the same for anti-DNA antibodies of HIV-infected and MS patients. Other sites are disposed mainly in the same protein sequences clusters of the H2B sequence, but they are different (Figure 6 and Figure 8; Table 1). This may also be due to the presence in the blood of HIV-infected and MS patients of various DNA-MBP complexes that differ significantly in additional components of these complexes. In such different complexes, different antigenic determinants of histones and MBP can be open or hooded.

Particular attention may be drawn to the fact that the number of sites for H2B hydrolysis by anti-DNA antibodies is much greater (31 sites) than its cleavage by antibodies against histones (7–12 sites) (Figure 6 and Figure 8; Table 1). Moreover, the sites of H2B hydrolysis by IgGs against DNA only partially coincide with those corresponding to the hydrolysis of H2B by IgGs against individual histones and MBP. Thus, IgGs against five different histones of HIV-infected and MS patients possess cross polyspecific complexation and enzymatic cross-reactivity. Moreover, antibodies with a high affinity for histones and MBP can efficiently hydrolyze DNA and vice versa. One of the most probable pathways from our point of view for the formation of such polyreactive antibodies can be the formation of binding sites and active centers of abzymes against the combined antigenic determinants of their protein and nucleic sequences. This pathway is supported by evidence that the active centers of monoclonal light chains of SLE antibodies can combine from two to three active centers of proteases and nucleases.

At the same time, as mentioned above, in human blood, milk, and placenta, IgGs to various antigens undergo extensive LH half-molecule exchange, leading to H_2_L_2_ IgGs containing HL halves variable fragments of which recognize different antigens [38,39,40,41]. Thus, it could not be ruled out that some intact IgGs may include HL halves against different antigens, for example, histones and MBP, histones and DNA, and MBP and histones. This pathway of antibody formation can also contribute to the formation of polyspecific abzymes exhibiting catalytic cross-reactivity.

## 4. Material and Methods

### 4.1. Chemicals, Donors, and Patients

All substances used, including electrophoretically homogeneous human H2B histone and an equimolar mixture of H2B, H2A, H1, H3, and H4, were purchased from Sigma (St. Louis, MO, USA). Protein G-Sepharose, BrCN-activated Sepharose, and Superdex 200 HR 10/30 were from GE Healthcare (GE Healthcare, New York, NY, USA). Human MBP was bought from the Center of Molecular Diagnostics and Therapy (DBRC, Moscow, Russia). Protein Sepharose columns containing immobilized MBP and individual histones or their mixture were obtained under the manufacturer’s protocol using BrCN-activated Sepharose, MBP, individual histones, or their mixture.

Proof that IgG preparations of MS and HIV-infected patients split five histones (H1-H4), MBP, and DNA was earlier attained using IgGs from the blood plasma of such sick patients [11,12,21,22,43,44]. Patient medical characteristics have been described earlier in [43,44]. Specialists of the Novosibirsk Medical University established the diagnosis of MS and HIV-infected patients. MS diagnosis was based on the classification of McDonald [64] and Kurtzke’s Expanded Disability Status Scale (EDSS) [65]. Blood plasma and IgGs of HIV-infected patients (16 at the stage of generalized lymphadenopathy and 13 at the stage of pre-AIDS) under the classification of Center of Disease Control and Prevention described in [43,44] were used. In all cases, the blood sampling protocols meet the guidelines of the human ethics hospital committee (Ethics committee of Novosibirsk State Medical University, Russia; number 105-HIV; 07, 2010). This committee approved this study following the guidelines of the Helsinki ethics committee, including a written agreement of patients to present their blood for scientific purposes. The MS patients at entry had no infectious symptoms. All patients gave legal written consent to provide their blood for scientific purposes. In this study, we used the IgG preparations described earlier [43,44].

### 4.2. Antibody Purification

Electrophoretically homogeneous IgGs preparations were isolated earlier from the blood plasma of 59 MS [44] and 32 HIV-infected [43] patients. All IgGs were first purified using affinity chromatography of the plasma components on Protein G-Sepharose and then additionally separated by gel FPLC gel filtration in acidic buffer (pH 2.6) using a Superdex 200 HR 10/30 column as in [43,44]. After 6–7 days of storage at 4 °C for refolding, the separated IgGs were used for different types of assays. In addition, IgGs were separated from any potential contaminations; they were filtered using 0.1 μm Millex filters. IgGs analysis for homogeneity was performed using SDS-PAGE: 5–17% gradient gels (0.1% SDS); all IgGs were visualized using silver staining [43,44,45,46,47,48,49,50,51].

Earlier, after SDS-PAGE of IgGs, cross-sections of the longitudinal gel slices (3–4-mm) were used to obtain eluates. In this work, we used preparations of IgG antibodies obtained and characterized earlier [43,44,45,46,47,48,49,50,51]. It was shown that all IgGs do not contaminate any canonical nucleases or proteinases [43,44]. Proteolytic and DNase activities of HIV-infected and MS patients were found only in the eluates in gel fragments corresponding to IgGs.

### 4.3. Chromatography of IgGs on Affinity Sorbents

To check the hypothesis about the possibility of cross-hydrolysis by any previously obtained IgG preparations hydrolyzing histones, MBP, and DNA, seven preparations of polyclonal IgGs with the maximum activity in the hydrolysis of histones, MBP, and DNA were selected for such an analysis from each group of the IgG preparations described above [43,44,45,46,47,48,49,50]. Two mixtures were obtained using these IgG preparations: ms-IgG_mix_ and hiv-IgG_mix_. To isolate IgGs against MBP, ms-IgG_mix_, and hiv-IgG_mix_ were first subjected to affinity chromatography on MBP-Sepharose. Nonspecifically bound antibodies were eluted from sorbent using 20 mM Tris-HCl buffer (pH 7.5) containing 0.2 M NaCl as in [43,44,45,46,47,48,49,50]. Anti-MBP IgGs were eluted first from sorbents with 3.0 M NaCl and then additionally using acidic buffer (pH 2.6). These fractions in each case were separately combined and dialyzed against 20 mM Tris-HCl buffer. To separate possible admixtures of IgGs against histones, the fractions obtained from ms-IgG_mix_ and hiv-IgG_mix_ were applied on the Sepharose (His5-Sepharose) containing five immobilized histones (H1–H4). The fractions obtained during the loading and washing of the column with 4 mL 20 mM Tris-HCl buffer were further used as anti-MBP ms-IgGs anti-MBP hiv-IgGs.

The fractions of IgGs of HIV-infected and MS patients were eluted from MBP-Sepharose at loading, and the column washing with 5 mL contained IgGs against five histones. It was subjected to one more passing through MBP-Sepharose and then used to isolate IgGs against five histones using His5-Sepharose. Nonspecifically bound IgGs were eluted with 0.2 M NaCl, while anti-histone IgGs used 3.0 M NaCl and acidic buffer (pH 2.6). A mixture of these two fractions was further used as anti-his-ms-IgGs anti-his-hiv-IgGs.

To obtain antibodies of MS patients against five individual histones, anti-his ms-IgGs fraction was applied first on H2B-Sepharose containing immobilized H2B histone. The fractions eluted at loading on H2B-Sepharose were applied on H1-Sepharose. Then, the fractions eluted when applied to the previous sorbent were applied to the next one: H2A-Sepharose, H3-Sepharose, and H4-Sepharose. All sequential chromatographies were performed similarly to that in the case of His5-Sepharose and MBP-and Sepharose. IgGs against five individual histones were specifically eluted from each sorbent using a buffer of pH 2.6. These fractions were designated as anti-H2B, anti-H1, anti-H2A, anti-H3, and anti-H4 ms-IgGs. The anti-his-hiv-IgGs preparation was used to obtain only antibodies against H2B histone by the method described above.

To obtain anti-DNA IgGs, the fractions with no affinity for MBP-Sepharose and then for His-5 Sepharose (eluted from these sorbents upon loading) were additionally passed twice through these two affinity sorbents. The final fractions were used to obtain anti-DNA antibodies of HIV-infected and MS patients. They were applied on DNA–cellulose (5 mL, equilibrated with 20 mM Tris-HCl buffer, pH 7.5). After elution of antibodies with a low affinity for DNA 0.2 M NaCl, anti-DNA IgGs were eluted first with 3.0 M NaCl and then with the acidic buffer. By combining these two factions, anti-DNA-ms-IgGs and anti-DNA-hiv-IgGs were obtained.

### 4.4. Protease Activity Assay

The reaction mixtures (10–20 μL) contain 24 mM Tris-HCl (pH 7.5), 0.8–1.0 mg/mL H2B or a mixture of histones, or 1.0 mg/mL MBP, and 0.01–0.15 mg/mL IgGs against five individual histones, DNA or MBP. Each mixture was incubated for 1–24 h at 37 °C. The efficiency of histones and MBP splitting was analyzed by SDS-PAGE (15% gels) in the absence of dithiothreitol under nonreducing conditions as in [45,46,47,48,49,50,51,52]. The products of protein degradation were detected using staining of gels with Coomassie Blue. The gels were scanning quantified as in [46,47,48,49,50,51] using Image Quant v5.2 software. The efficiency of protein hydrolysis was evaluated from the reduction in the starting proteins’ content compared to the control–incubation of histones or MBP in the absence of IgGs.

### 4.5. DNA Hydrolysis

The reaction mixtures (10–15 μL) contained: 4 mM MgCl_2_, 0.2 mM CaCl_2_, 20 mM Tris-HCl, pH 7.5, 10 μg/mL supercoiled (sc)DNA *pBluescript* plasmid, and 2–30 μg/mL IgGs, similar to [11,12]. After incubation for 1.0–4.0 h at 37 °C, 2.5 μL of loading buffer containing 1% SDS, 50 mM EDTA, pH 8.0, 30% glycerol and 0.005% bromophenol blue was added to the reaction mixture. Electrophoresis was performed using 0.8% agarose gel until the bromophenol blue migrated 2/3 of the way. ScDNA in the gel was stained with a solution of ethidium bromide (0.5 μg/mL, 1–2 min). Using an orange filter, the gel was imaged using a Gel Doc gel documentation system from Bio-Rad (Bio-RAD, Berkeley, CA, USA). The photographs of the gels were counted using the ImageQuant 5.2 program. The level of activity of IgG preparations was determined by the degree of hydrolysis of the scDNA form of the plasmid to relaxed-form DNA; the linear harts of the dependencies taking into account its hydrolysis in the absence of antibodies were used.

### 4.6. MALDI-TOF Analysis of Histones Hydrolysis

H2B hydrolysis by IgGs against five individual histones, MBP, and DNA was performed using the Reflex III system (Bruker Company, Frankfurt, Germany): VSL-337 ND nitrogen 337 nm laser with a 3 ns pulse duration as in [45,46,47,48,49,50,51]. Mixtures (12–14 µL) containing 20 mM Tris-HCl buffer (pH 7.5), 0.9 mg/mL H2B histone and 0.03–0.05 mg/mL one of IgGs were incubated at 30 °C for 0–20 h. To 1.4 µL of the sinapinic acid matrix mixed with 1.4 µL of 0.2% trifluoroacetic acid, 1.4 µL of the solutions containing one of the histones before or after incubation with different IgGs against five individual histones was added; 1.2–1.3 µL of these mixtures after loading on the MALDI plates was air-dried. Every MALDI-TOFF mass spectrum was calibrated using Bruker Standard Daltonic protein mixtures II and I (Bruker, Frankfurt, Germany) in two internal/or external calibration modes. The analysis of hydrolysis products’ molecular masses corresponding to specific sites of histones degradation by IgG preparations against histones, MBP, or DNA was carried out using Protein Calculator v3.3 (Scripps Research Institute; La Jolla, CA, USA).

### 4.7. Analysis of Sequence Homology

The analysis of sequence homology between histones and MBP was performed using *the lalign* site (http://www.ch.embnet.org/software/LALIGN_form.html, accessed on 1 January 2008).

### 4.8. Statistical Analysis

The results are presented using mean ± S.D. of 7–9 independent MALDI mass spectra for each sample of every histone hydrolysis by IgGs against all histones and MBP.

## 5. Conclusions

In this work, for the first time, we analyzed the possible pathways for the formation of antibodies and abzymes that exhibit offbeat polyspecificity in recognition during complex formation and the possibility of catalysis by the same antibody of several reactions. Using IgGs against five histones, myelin basic protein, and DNA, it was first shown that abzymes against each of these antigens effectively hydrolyze all three antigens. It was suggested that the formation of abzymes, whose active centers can catalyze several reactions, can occur mainly in two ways. On the one hand, such abzymes can be antibodies against hybrid antigenic determinants consisting of protein and nucleic sequences. Another method of their formation may be associated with the previously described phenomenon of IgGs’ extensive LH half-molecule exchange, leading to H_2_L_2_ molecules containing HL halves with variable fragments recognizing different antigens.

## Figures and Tables

**Figure 1 ijms-23-08102-f001:**
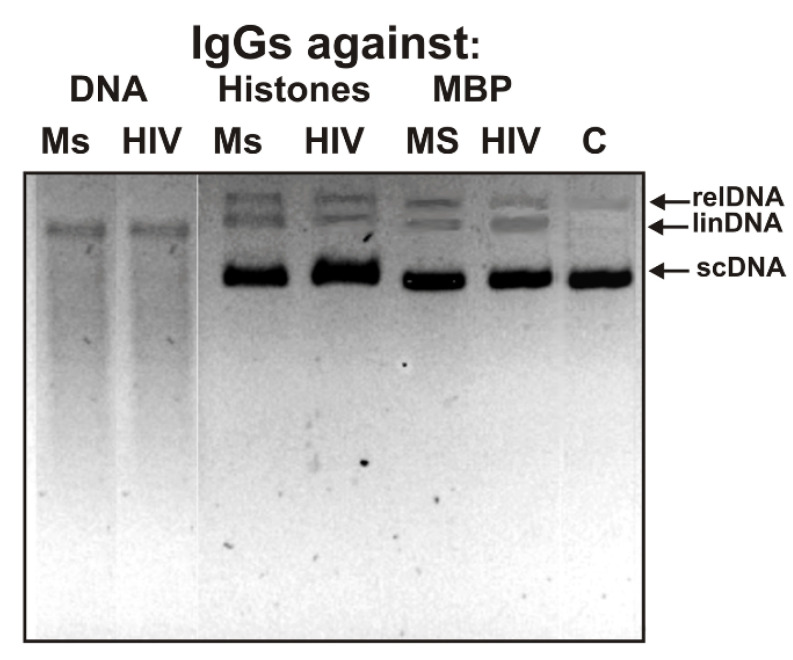
Analysis of the relative activity of IgGs against DNA, five different histones, and MBP in the hydrolysis of supercoiled DNA for 3 h from MS and HIV-infected patients. Designations of IgGs and diseases are shown in the figure: supercoiled DNA (scDNA), relaxed DNA (relDNA containing one break), linear DNA (linDNA containing several breaks).

**Figure 2 ijms-23-08102-f002:**
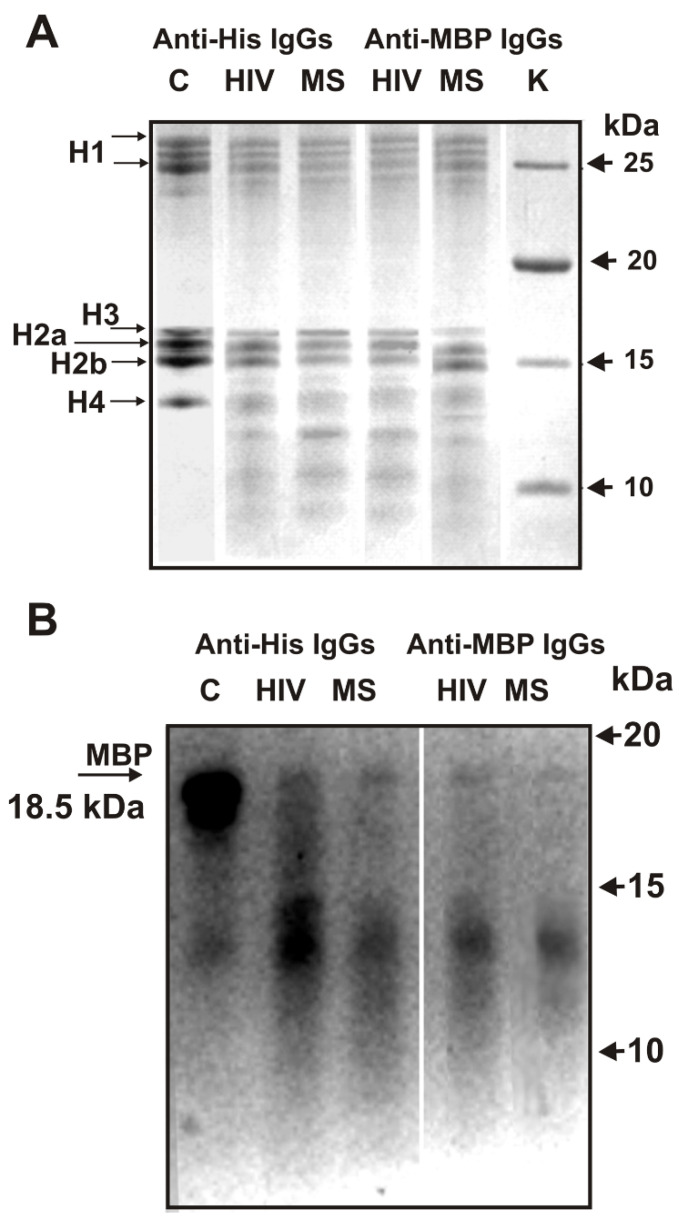
SDS-PAGE analysis of hydrolysis of five histones (**A**) and MBP (**B**) by antibodies against five histones and MBP from the blood of HIV-infected and MS patients. Lanes C corresponds to the histones (**A**) and MBP (**B**) incubated without IgGs. Lane K–control markers of proteins molecular mass (**A**). MBP and a mixture of five histones with and without IgGs (0.03 mg/mL) were incubated for 12 h. Designations of IgGs and diseases are shown in the figure.

**Figure 3 ijms-23-08102-f003:**
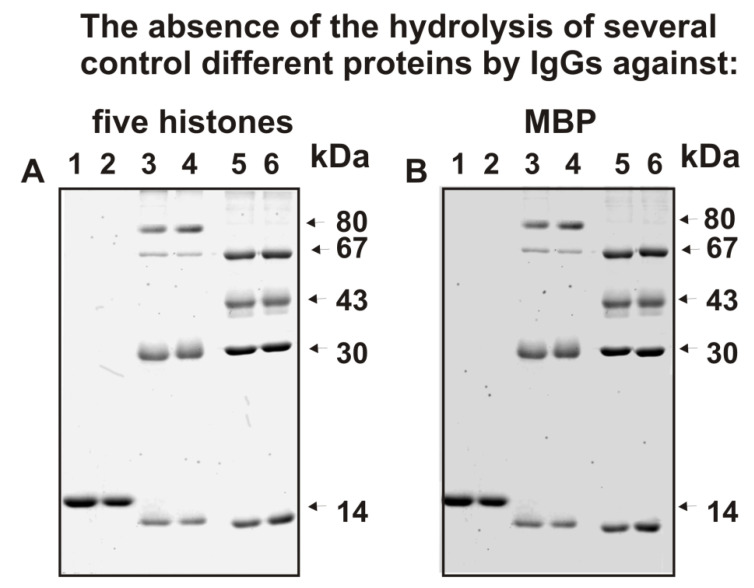
SDS-PAGE analysis of hydrolysis by IgGs of different control proteins (**A**,**B**); odd and even numbers of line pairs correspond to various proteins incubated with and without IgG preparation, respectively: 1 and 2–egg lysozyme (~14 kDa), 3 and 4–mixture of human milk lactoferrin (~80 kDa), human albumin (~67 kDa), human casein (~30 kDa), and human lactalbumin (~14 kDa); 5 and 6–mixture of some protein molecular mass markers –bovine albumin (~67 kDa), egg ovalbumin (~43 kDa), bovine carbonic anhydrase (~30 kDa), and bovine milk α-lactalbumin (~14.4 kDa). All mixtures with and without IgGs (0.03 mg/mL) were incubated for 12 h.

**Figure 4 ijms-23-08102-f004:**
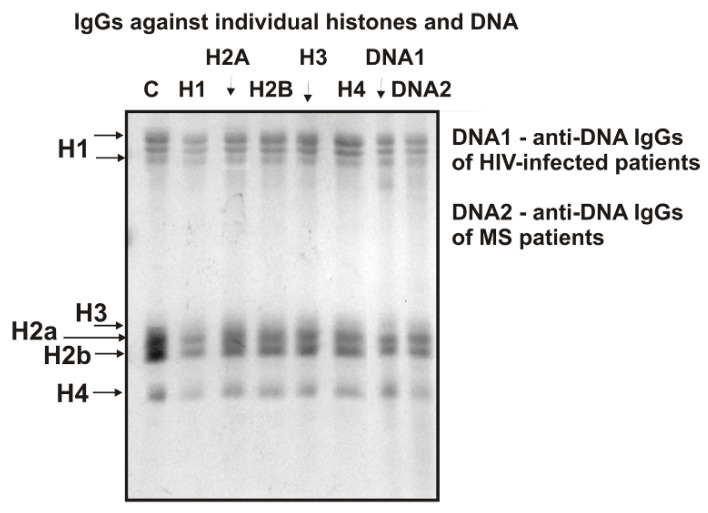
SDS-PAGE analysis of hydrolysis of five histones by IgGs against five individual histones from the blood of MS patients and anti-DNA IgGs of HIV-infected and MS patients. Lane C corresponds to the histones incubated without IgGs. The mixture of five histones with IgGs (0.03 mg/mL) and without Abs were incubated for 12 h. Designations of IgGs and diseases are shown in the figure.

**Figure 5 ijms-23-08102-f005:**
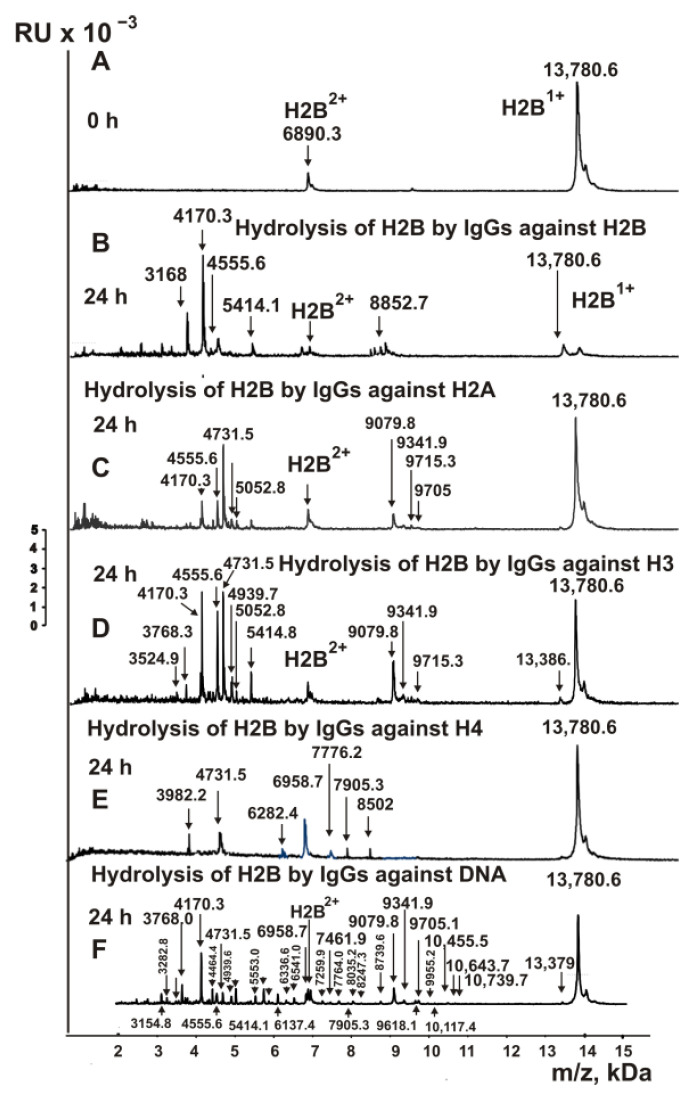
MALDI spectra show H2B histone (0.8 mg/mL) over time hydrolysis (0–24 h) in the presence of IgGs (0.045 mg/mL) against H2B, H2A, H3, H4 and DNA (**A**–**F**). All used IgG preparations and m/z of hydrolysis products (Da) are indicated on the panels.

**Figure 6 ijms-23-08102-f006:**
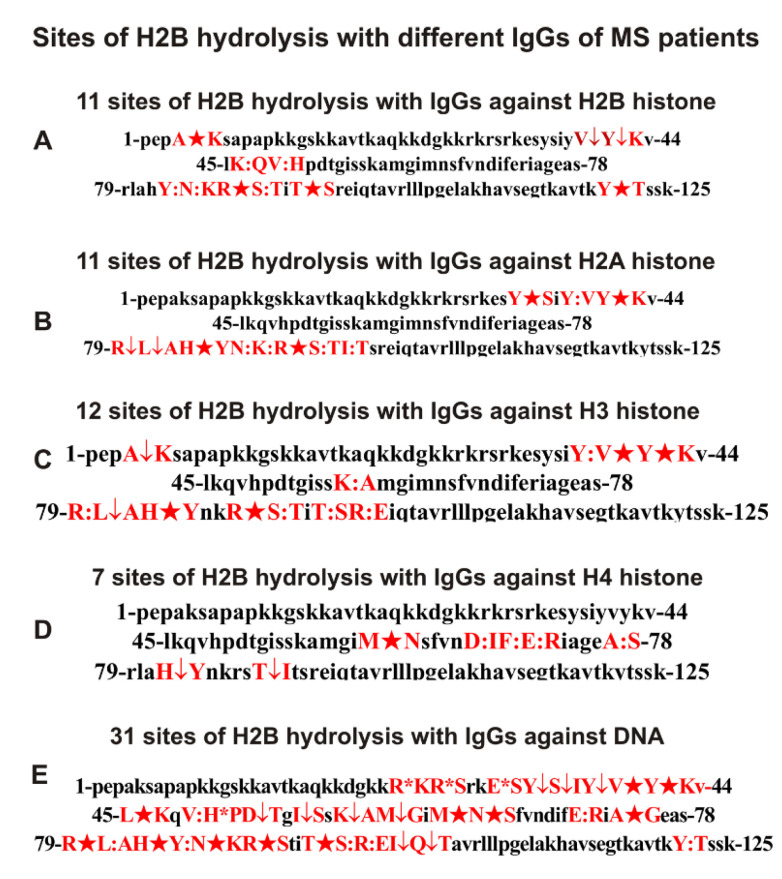
Sites of H2B hydrolysis by IgGs of MS patients against H2B (**A**), H2A (**B**), H3 (**C**), H4 (**D**), and DNA (**E**). Major sites of H2B cleavage are marked by big stars (★), moderate ones by arrows (↓), and minor sites of the splitting by colons (:) (**A**–**E**). Clusters and sites of histone hydrolysis are shown in red.

**Figure 7 ijms-23-08102-f007:**
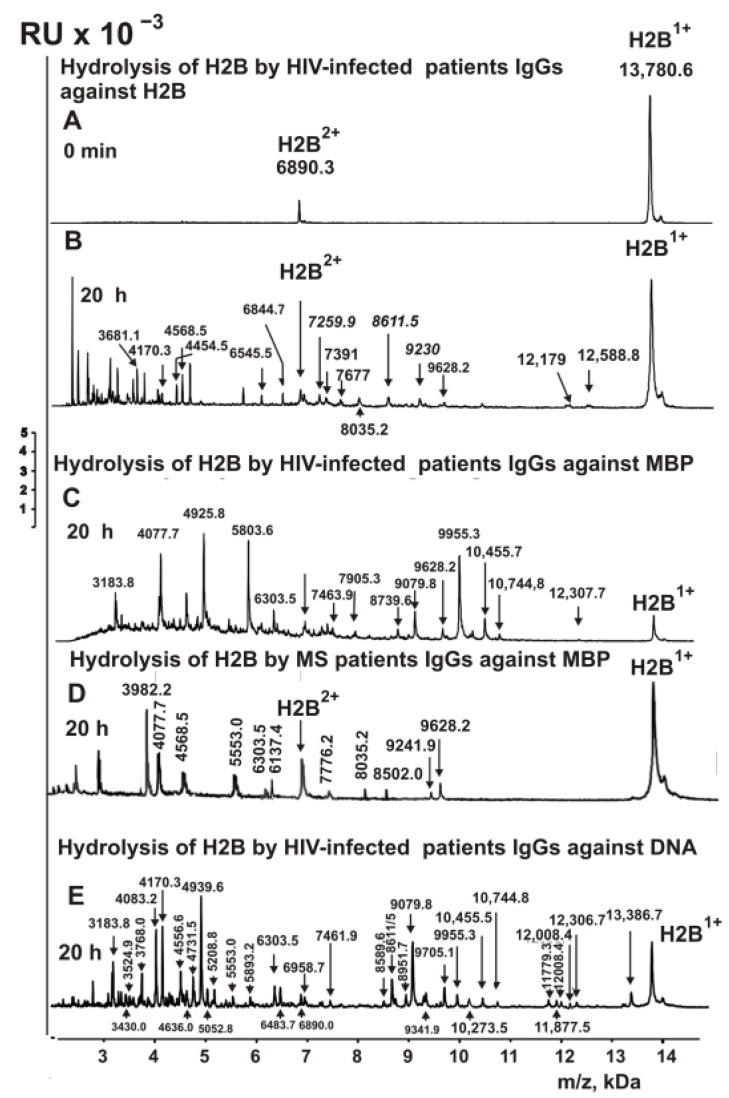
MALDI spectra show H2B histone (0.8 mg/mL) over time hydrolysis (0–20 h) in the absence (**A**) and in the presence of HIV-infected patients IgGs (0.045 mg/mL) against H2B (**B**), MBP (**C**) and DNA (**E**). (**D**) shows H2B histone cleavage with MS patients IgGs against MBP. All used IgG preparations, and m/z of hydrolysis products (Da) are indicated on the panels.

**Figure 8 ijms-23-08102-f008:**
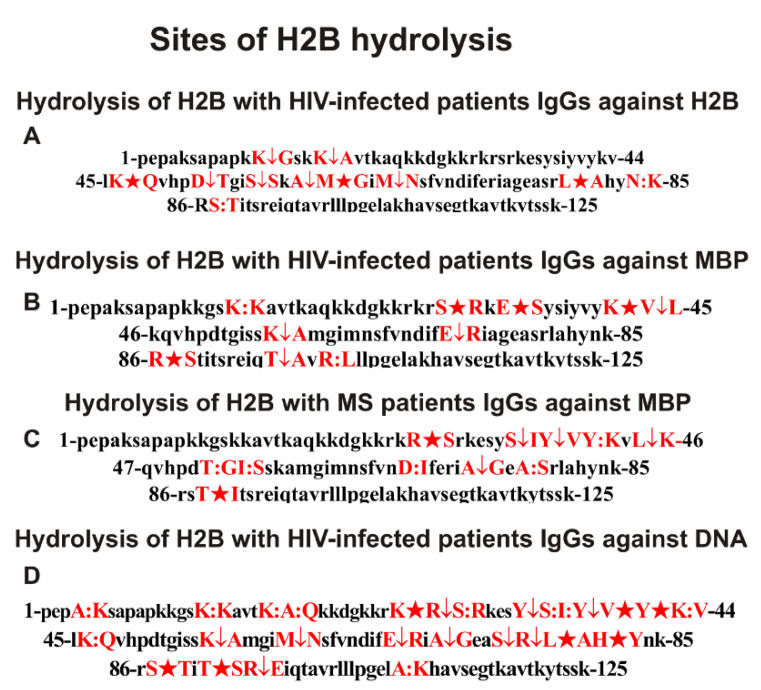
Sites of H2B hydrolysis by IgGs of HIV-infected patients against H2B (**A**), MBP (**B**), and DNA (**D**), as well as with Abs of MS patients against MBP (**C**). Major sites of H2B cleavage are marked by big stars (★), moderate ones by arrows (↓), and minor sites of the splitting by colons (:) (**A**–**D**). Clusters and sites of histone hydrolysis are shown in red.

**Table 1 ijms-23-08102-t001:** Sites of H2B histone hydrolysis by MS and HIV-infected patients IgGs against five histones, MBP, and DNA *.

MS Patients, Type of IgGs						HIV-Infected Patients, Type of IgGs		
Anti-H2B	Anti-H2A	Anti-H3	Anti-H4	Anti-MBP	Anti-DNA	Anti-H2B	Anti-MBP	Anti-DNA
11 Sites	11 Sites	12 Sites	7 Sites	10 Sites	31 Sites	11 Sites	10 Sites	27 Sites
-	-	-	-	-	**R29-K30**	-	-	-
**4A-5K**	-	* A4-K5 *	-	-	-	-	-	4A-5K
-	-	-	-	-	-	* **12K-13G** *	-	-
-	-	-	-	-	-	-	15K-16K	15K-16K
-	-	-	-	-	-	* **16K-17A** *	-	-
-	-	-	-	-	-	-	-	K20-A21
-	-	-	-	-	-	-	-	A21-Q22
-	-	-	-	-	-	-	-	K30-R31
-	-	-	-	R31-S32	**R31-S32**	-	-	R31-S32
-	-	-	-	-	-	-	S32-R33	**S32-R33**
-	-	-	-	* **S38-I39** *	**E35-S36**	-	E35-S36	-
-	Y37-S38	-	-	-	* **Y37-S38** *	-	-	* **Y37-S38** *
-	-	-	-	-	* **S38-I39** *	-	-	**S38-I39**
-	-	-	-	-	-	-	-	**I39-Y40**
-	Y40-V41	Y40-V41	-	* **Y40-V41** *	* **Y40-V41** *	-	-	* **Y40-V41** *
* **V41-Y42** *	-	**V41-Y42**	-	**Y42-K43**	V41-Y42			** V41-Y42 **
* **Y42-K43** *	Y42-K43	Y42-K43	-	-	**Y42-K43**	-	-	**Y42-K43**
-	-	-	-	-	-	-	**K43-V44**	**K43-V44**
-	-	-	-	-	-	-	* **V44-L45** *	-
-	-	-	-	* **L45-K46** *	**L45-K46**	-	-	-
**K46-Q47**	-	-	-	-	-	**K46-Q47**		**K46-Q47**
**V48-H49**	-	-	-	-	**V48-H49**	-	-	-
** - **	-	-	-	-	**H49-P50**	-	-	-
-	-	-	-	-	* **D51-T52** *	* **D51-T52** *	-	-
-	-	-	-	**T52-G53**	-	-	-	-
-	-	-	-	**I54-S55**	* **I54-S55** *	-	-	-
-	-	-	-	-	-	* **S55-S56** *		
-	-	**K57-A58**	-	-	* **K57-A58** *	-	* **K57-A58** *	* **K57-A58** *
-	-	-	-	-	-	* **A58-M59** *	-	-
-	-			-	* **M59-G60** *	** *M59-G60* **	-	-
-	-	-	M62-N63		**M62-N63**	* **M62-N63** *	-	* **M62-N63** *
-	-	-	-	-	**N63-S54**	-	-	-
-	-	-	**D68-I69**	**D68-I69**	-	-	-	-
-	-	-	**F70-E71**	-	-	-	-	-
-	-	-	**E71-R72**	-	**E71-R72**	-	**E71-R72**	**E71-R72**
-	-	-	-	* **A74-G75** * ** **	**A74-G75**	-	-	* **A74-G75** *
-	-	-	**A77-S78**	**A77-S78**				
-	-	-	-	-	-	-	-	* **S78-R79** * ** **
-	** * R79-L80 * **	**R79-L80**	-	-	**R79-L80**	-	-	* **R79-L80** *
-	** * L80-A81 * **	** * L80-A81 * **	-	-	**L80-A81**	**L80-A81**	-	**L80-A81**
-	H82-Y83	H82-Y83	** * H82-Y83 * **		H82-Y83	-	-	H82-Y83
**Y83-N84**	** * - * **	** * - * **	-	-	**Y83-N84**	-	-	-
**N84-K85**	**N84-K85**	-	** * - * **	-	**N84-K85**	**N84-K85**	-	-
	**K85-R86**	-	-	-	-	-	-	-
**R86-S87**	R86-S87	R86-S87	-	-	**R86-S87**	-	**R86-S87**	-
**S87-T88**	**S87-T88**	**S87-T88**	-	* - *	-	S87-T88	-	S87-T88
-	-	-	* **T88-I89** *	T88-I89	-	-	-	-
-	**I89-T90**	-	-	-	-	-	-	-
	-	-	-	-	**T90-S91**	-	-	**T90-S91**
-	-	-	** * - * **	-	-	-	-	-
**T90-S91**	-	**T90-S91**	-	-	-	-	-	-
-	-	-	-	-	**S91-R92**	-	-	-
-	-	**R92-E93**	-	-	**R92-E93**			* **R92-E93** *
-	-	-	-	-	* **I94-Q95** *	-	-	-
-	-	-	-	-	* **Q95-T96** *	-	-	-
-	-	-	-	-	-	-	* **T96-A97** *	
-	-	-	-	-	-		**R99-L100**	
								**A107-K108**
**Y121-T122**	-	-	-	-	**Y121-T122**			

* Major hydrolysis sites are marked in bold (red), moderate in italics (green) and minor sites in bold (black): missing.

## Data Availability

The data supporting our study results are included in the article.

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
