# Peer review of "Autoimmune Diseases: Enzymatic cross Recognition and Hydrolysis of H2B Histone, Myelin Basic Protein, and DNA by IgGs against These Antigens"

_ijms, 2022, doi:10.3390/ijms23158102_

Round 1

Reviewer 1 Report

The authors have written an interesting article showing how antibodies with catalytic activities (abzymes) hydrolyze various autoantigens (Abs are provided by patients with MS and HIV).

However, I am not quite sure what they meant by that title,...which ADs?..SLE is only mentioned..what about others? is this a fusion of two recently published articles? maybe better to provide a review about this theme now. plus it is not quite in the spirit of English,

However, my biggest concern is previously published articles with similar results by the same authors:

1. "HIV-Infected Patients: Cross Site-Specific Hydrolysis of H2a and H2b Histones and Myelin Basic Protein with Antibodies against These Three Proteins," from 2020th (Biomolecules MDPI).

2. "Multiple Sclerosis: Enzymatic Cross Site-Specific Hydrolysis of H1 Histone by IgGs against H1, H2A, H2B, H3, H4 Histones, and Myelin Basic Protein," from 2021.(MDPI BIomolecules).

My main comment is:

Since I do not see much difference between the published articles and this submitted article, I ask the authors to state point by point all relevant differences between these two previously published articles (mentioned above) and the article submitted here to IJMS (separately methods and results).

Author Response

The authors have written an interesting article showing how antibodies with catalytic activities (abzymes) hydrolyze various autoantigens (Abs are provided by patients with MS and HIV).

However, I am not quite sure what they meant by that title,...which ADs?..SLE is only mentioned..what about others? is this a fusion of two recently published articles? maybe better to provide a review about this theme now. plus it is not quite in the spirit of English,

However, my biggest concern is previously published articles with similar results by the same authors:

  1. "HIV-Infected Patients: Cross Site-Specific Hydrolysis of H2a and H2b Histones and Myelin Basic Protein with Antibodies against These Three Proteins," from 2020th (Biomolecules MDPI).

    2. "Multiple Sclerosis: Enzymatic Cross Site-Specific Hydrolysis of H1 Histone by IgGs against H1, H2A, H2B, H3, H4 Histones, and Myelin Basic Protein," from 2021.(MDPI BIomolecules).

    My main comment is:

    Since I do not see much difference between the published articles and this submitted article, I ask the authors to state point by point all relevant differences between these two previously published articles (mentioned above) and the article submitted here to IJMS (separately methods and results).

Answer:

Sorry, but this is original research paper

As for the methods, they are mostly close to those developed and used by us earlier in previously published works. However, in this work, we have developed a method for obtaining anti-DNA antibodies with high affinity for DNA and hydrolyzing H2B histone at other sites compared to anti-MBP and anti-histone antibodies.

Sorry, in article ["HIV-Infected Patients: Cross Site-Specific Hydrolysis of H2a and H2b Histones and Myelin Basic Protein with Antibodies against These Three Proteins"] it was shown that antibodies from the blood of HIV-Infected Patients hydrolyze myelin basic protein H2a and H2b Histones. In this[, article there is not a word about the hydrolysis of histones and DNA by antibodies from HIV-Infected patients and even more nothing about the polyspecificity of binding and catalytic cross-reactivity of antibodies with respect to H2B histone all antibodies against hive histones and especially DNA. Hydrolysis of H2B by anti-DNA antibodies of HIV-infected patients was carried out for the first time in this work.

In our article ["Multiple Sclerosis: Enzymatic Cross Site-Specific Hydrolysis of H1 Histone by IgGs against H1, H2A, H2B, H3, H4 Histones, and Myelin Basic Protein] there is data on hydrolysis by antibodies against MBP directly to MBP and antibodies against H1 by IgGs against H1, H2A, H2B, H3, H4 Histones. However, in this article there was no data on the hydrolysis of  H2b by antibodies against all five H1, H2A, H2b, H3, H4 Histones from patients with multiple sclerosis. Moreover, in this new article, using the example of H2B histone, it was shown for the first time that antibodies against each of the five histones effectively hydrolyze H2B histone and MBP, and antibodies against MBP also effectively hydrolyze H2B. The most original and unexpected result was found in the case of anti-DNA antibodies. It was shown for the first time that anti-DNA antibodies hydrolyze not only DNA, but also the individual H2B histone and MBP. This is a completely new and original result that anti-DNA antibodies effectively hydrolyze not only DNA, but also H2B and MBP, while antibodies against all histones and MBP hydrolyze DNA. This is completely new data on the possibilities of combining several sites of specific recognition of different antigens in one active site; in Abs having high affinity for protein or DNA (anti protein and anti-DNA antibodies) at least two active sites  with protease and DNase activities may be present. The fact that antibodies against H2B histone and against MBP can hydrolyze not only these proteins, but also DNA, while antibodies against DNA can split not only DNA but also these proteins is a completely new and somewhat unexpected result.

However, there is a reasonable explanation for this fact. for the first time, an explanation was given of how antibodies with extended multifunctionality of complexation and catalytic cross-reactivity can be formed. A reasonable explanation of these facts is given for the first time. Since complexes of five histones and DNA enter the blood during apoptosis of cells, antibodies can be formed not only against each of five individual histones, but also against antigenic determinants formed by different combinations of two or even three histones at their junction. In a histone-DNA complex, antibodies can be formed directly against DNA, as well as DNA fragments and protein sequences of each of the five histones and DNA with several histones.

The data of this work provide a completely new insight into the possible ways of formation of antibodies with functional polyreactivity of complex formation and catalytic cross-reactivity. This kind of data has not yet been in our previous articles. Our article provides a first powerful extension in understanding of the additional earlier not described capabilities of the immune human system.

Any generalization on the totality of the data obtained seems premature. It is very important to investigate whether the same situation exists in the polyreactivity of antibodies against histones, MBP and DNA in the case of hydrolysis of other histones – H2a, H3, and H4. And only then the general situation will be clear how a large number of antibodies harmful to humans with extended polyreactivity and catalytic cross-reactivity can be formed.

Thanks a lot for comments

Sincerely

Reviewer 2 Report

The manuscript entitled ” Autoimmune diseases: enzymatic cross recognition and hydrolysis of H2B histone, myelin basic protein, and DNA by IgGs against these antigens” by Georgy A. Nevinsky et al focuses on the analysis of the possible pathways for the formation of antibodies and abzymes that exhibit offbeat polyspecificity in recognition during complex formation and the possibility of catalysis by the same antibody of several reactions. Using IgGs against five histones, myelin basic protein, and DNA, it was first shown that abzymes against each of these antigens effectively hydrolyze all three antigens. The authors of the work conclude that the formation of abzymes, whose active centers can catalyze several reactions, can occur mainly in two ways.

The introduction of the article provides sufficient background and include all relevant references. The research design is appropriate. The used methods are adequately described and the results obtained are clearly presented enough. Besides, the conclusions of the authors are supported by the resalts exhibited. 

The work is very important and can be used as a basis for further research.

However, the following improvements of the article would be useful for a better understanding of the results obtained:

1. In the abstract the authors use the abbreviations ”LH half-molecule”, “to H2L2 molecules” but do not explain what they mean. 

2. Illustration 1 going with the article indicates the activity of IgGs against DNA, but the meaning of the abbreviations of the rel-, lin- and scDNA is not indicated.

3. Illustration 3 shows the data on the absence of hydrolysis by antibodies against five histones and MBP of eight control proteins, but there are only six bands (indicated) in the illustration/ but the illustration shows/indicates only six bands. Besides, for this illustration it is enough to show only one legend.

4. It is necessary to standardise all the legends.

5. It is necessary to describe of analogical studies by other scientist. 

Besides, 70 % of references are self-citations. 

Author Response

The manuscript entitled ” Autoimmune diseases: enzymatic cross recognition and hydrolysis of H2B histone, myelin basic protein, and DNA by IgGs against these antigens” by Georgy A. Nevinsky et al focuses on the analysis of the possible pathways for the formation of antibodies and abzymes that exhibit offbeat polyspecificity in recognition during complex formation and the possibility of catalysis by the same antibody of several reactions. Using IgGs against five histones, myelin basic protein, and DNA, it was first shown that abzymes against each of these antigens effectively hydrolyze all three antigens. The authors of the work conclude that the formation of abzymes, whose active centers can catalyze several reactions, can occur mainly in two ways.

The introduction of the article provides sufficient background and include all relevant references. The research design is appropriate. The used methods are adequately described and the results obtained are clearly presented enough. Besides, the conclusions of the authors are supported by the resalts exhibited. 

The work is very important and can be used as a basis for further research.

However, the following improvements of the article would be useful for a better understanding of the results obtained:

  1. In the abstract the authors use the abbreviations ”LH half-molecule”, “to H2L2 molecules” but do not explain what they mean. 

Answer:

It was done

Their formation may also be associated with the previously described phenomenon of IgGs extensive LH half-molecule (containing one L-light and one H-heavy chains) exchange leading to H2L2 molecules containing HL halves with variable fragments recognizing different antigens.

  1. Illustration 1 going with the article indicates the activity of IgGs against DNA, but the meaning of the abbreviations of the rel-, lin- and scDNA is not indicated.

Answer:

It was done

Analysis of the relative activity of IgGs against DNA, five different histones, and MBP in the hydrolysis of supercoiled DNA for 3 h from MS and HIV-infected patients. Designations of IgGs and diseases are shown in the Figure: supercoiled DNA (scDNA), relaxed DNA (relDNA containing one break), linear DNA (linDNA containing several breaks).  

  1. Illustration 3 shows the data on the absence of hydrolysis by antibodies against five histones and MBP of eight control proteins, but there are only six bands (indicated) in the illustration/ but the illustration shows/indicates only six bands. Besides, for this illustration it is enough to show only one legend.

Answer:

Sorry, there are actually 6 bands, but some bands contain several control proteins

Lanes 1 and 2 - 1 protein – egg lysozyme (~ 14 kDa )

Lanes 3 – 4 correspond to four  proteins  – mixture of human milk lactoferrin (~80 kDa), human albumin (~67 kDa), human casein (~30 kDa), and human lactalbumin (~14 kDa);

Lanes 5 and 6 – correspond to four  proteins ; mixture of some protein molecular mass markers –bovine albumin (~67 kDa), egg ovalbumin (~43 kDa), bovine carbonic anhydrase (~30 kDa), and bovine milk α-lactalbumin (~14.4 kDa).

Thus, there are nine bands, but the same protein (human albumin; ~67 kDa) )  is present in two bands -3 – 4 and 5-6. Finally we used 8 proteins

  1. It is necessary to standardise all the legends.

Answer:

All signatures are standardized

  1. It is necessary to describe of analogical studies by other scientist. 

Answer:

Sorry, but this is not yet possible, since there are no studies of this kind in the literature yet.

Besides, 70 % of references are self-citations. 

Answer:

We would be happy to provide links to articles by other authors. But at present, there are no studies in the literature of a detailed study of natural abzymes by other authors. If we do not provide links to your previous publications, then it will not be clear what is new sense of this article showing of a more detailed and in-depth study based on data from previous publications.

Thanks a lot for comments

Sincerely

Prof. Georgy A. Nevinsky

Round 2

Reviewer 1 Report

The authors have given all the necessary answers in accordance with the request.